# Look-Ahead Selective Plasticity for Continual Learning of Visual Tasks

## Abstract

Contrastive representation learning has emerged as a promising technique for continual learning as it can learn representations that are robust to catastrophic forgetting and generalize well to unseen future tasks. Previous work in continual learning has addressed forgetting by using previous task data and trained models. Inspired by event models created and updated in the brain, we propose a new mechanism that takes place during task boundaries, i.e., when one task finishes and another starts. By observing the redundancy-inducing ability of contrastive loss on the output of a neural network, our method leverages the first few samples of the new task to identify and retain parameters contributing most to the transfer ability of the neural network, freeing up the remaining parts of the network to learn new features. We evaluate the proposed methods on benchmark computer vision datasets including CIFAR10 and TinyImagenet and demonstrate state-of-the-art performance in the task-incremental, class-incremental, and domain-incremental continual learning scenarios. [1]

## 1 Introduction

Deep neural networks (DNNs) have been solving a variety of computer vision tasks with high performance. While this feat has been achieved via access to large and diverse datasets, in many practical scenarios data is not available in its entirety at first and becomes available over time, potentially including new unseen classes and different target distributions. When presented with a sequence of classification tasks to learn and remember, DNNs suffer from a well-known catastrophic forgetting problem (McCloskey & Cohen, 1989), losing their performance on previous classification datasets abruptly. To address this issue, various continual learning algorithms have been proposed. A class of methods introduces a way of identifying and retaining parameters most important for the performance of the network on previous tasks. Retention of these parameters is usually done by either regularization or freezing (parameter isolation). Methods such as Elastic Weight Consolidation (EWC) (Kirkpatrick et al., 2017), Synaptic Intelligence (SI) (Zenke et al., 2017), PackNet (Mallya & Lazebnik, 2018), and one similar to our work, Attention-based Structural Plasticity (Kolouri et al., 2019) belong to this class of continual learning approaches. Another class of methods such as Learning without Forgetting (LwF) (Li & Hoiem, 2017) attempt to retain performance on previous tasks via storing the networks trained for previous tasks and using their output to define a distillation loss on the output of the network being trained on the current task. Rehearsal is another approach to mitigate forgetting widely used in continual learning. Rehearsal-based methods such as iCaRL (Rebuffi et al., 2017), GSS (Aljundi et al., 2019), GEM (Lopez-Paz & Ranzato, 2017), and CLS-ER (Arani et al., 2022) store a small number of training samples from previous tasks in a memory, while other methods such as (Shin et al., 2017; Seff et al., 2017) train a generative model to produce training samples similar to previous tasks. During the training of the current task, the network is concurrently trained on the current task samples as well as samples from the memory. There have also been promising meta-learning approaches to continual learning as in Meta Experience Replay (MER) (Riemer et al., 2018), Online Meta-Learning (OML) (Javed & White, 2019), the Neuromodulated Meta-Learning Algorithm (ANML) (Beaulieu et al., 2020), and La-MAML Gupta et al., 2020.

While the aforementioned continual learning methods are successful to some extent in mitigating "forgetting", whether or not the regularization or isolation of parameters, distillation, or meta-

---

[1]The source code is provided in the supplementary materials.

learning will help in learning new unseen tasks is not clear. In fact, in regularization and parameter isolation approaches, parameters are identified as important by some forms of evaluation on past tasks, without attention to whether these parameters will transfer to future tasks. Similarly, rehearsal-based approaches rely on some forms of regularization or gradient alignment with regard to past task data to achieve good performance. While recent work Wang et al., 2021; Lin et al., 2022 considers features learned from new task data, they do not encourage learning of features that generalize to all tasks seen so far and are more likely to transfer. Likewise, recent meta-learning approaches like La-MAML (Gupta et al., 2020) use gradient-alignment heuristics to modulate the plasticity of parameters but pay little attention to redundancy and the contribution of parameters to generalizability while being computationally expensive compared to other continual learning methods. Thus, there has been a general lack of attention regarding the transfer of continually learned knowledge to future tasks. A recent approach named $Co^2L$ (Cha et al., 2021) questioned whether preserved past knowledge generalizes to future tasks and observed that contrastively learned representations (Chen et al., 2020; Khosla et al., 2020) transfer better and forget less, compared to learning based on the cross entropy loss.

Aiming for a continual learning approach that mitigates forgetting while learning representations that transfer well to unseen data, we were inspired to build upon the contrastive learning framework (Cha et al., 2021; Khosla et al., 2020). In contrastive learning, we will be working with an encoder mapping input images to vectors (*representations*), a projection head mapping representations to vectors (named *embeddings*) on which a contrastive loss is defined, and a decoder (linear transformation) mapping the extracted representations to class probabilities at inference time. We build our approach around $Co^2L$ (Cha et al., 2021), but importantly, we will be regularizing produced embeddings and network parameters selectively, based on how likely they are to transfer to future tasks. In doing so, we revisit assumptions on access to data at each point in time and outline our inspirations from Event Models theorized to enable update of context representations in the brain.

**Task Boundaries and Event Models:** Events are how we understand the world around us. While the world seems to be a continuous stream of twists and turns, evidence suggests we perceive it as discrete events in various spatiotemporal scales (Radvansky & Zacks, 2011; Stawarczyk et al., 2021; Zacks et al., 2007). The brain has been theorized to operate and make sense of the world by updating and maintaining representations of the current situation, also known as *Event Models* (Radvansky & Zacks, 2011; Stawarczyk et al., 2021; Zacks et al., 2007). Inspiring to our work, event models are believed to be updated mainly at *event boundaries* (Radvansky & Zacks, 2011; Stawarczyk et al., 2021; Zacks et al., 2007). These boundaries are thought to be detected by a rise in perceptual prediction error, i.e., when the brain's visual model makes predictions of the world that start to diverge from what is actually happening (Radvansky & Zacks, 2011; Stawarczyk et al., 2021; Zacks et al., 2007). Interestingly, the said boundaries also exist in the field of continual learning at the moment where the first batch of new task data arrives (or any point in time where the model's prediction accuracy drops significantly). We will refer to these boundaries as *task boundaries*. While performing various kinds of computation during task boundaries is not new in continual learning, methods that do such computations (e.g., EWC (Kirkpatrick et al., 2017)) do not make use of all the information available at task boundaries. In the specific case of EWC (Kirkpatrick et al., 2017), a regularization strength for each network parameter is calculated using the previous task data, with no attention to the first batch of data coming from the new task. Until now, continual learning approaches have been focused on using past task data and models to overcome forgetting. Assuming a stream data where the data distribution changes, we can mark any time the model's performance on a batch of data drops as a task boundary and assign data before this batch to the previous task. Consequently, the batch of data the model did not perform well will belong to a new task. Accessing this batch of data is not a new assumption as we are merely choosing to delay the application of continual learning methods until after the first batch of new task data is received, rather than applying them after the last batch of the previous task. This way, we also give up access to the last batch of data from the previous task. Overall, we assume access to one batch of data, some samples in a memory, and a snapshot of the model taken when training on the new task started, same as previous work (Cha et al., 2021).

**Redundancy in Contrastive Learning:** Recent work suggests that most continual learning methods favor stability over plasticity, that is, they focus on not forgetting about past tasks by preserving learned parameters and sacrificing flexibility to learn new knowledge (Kim & Han, 2023). It is thus favorable to introduce less regularization into continual learning methods by only retaining

parts of the learned network that are vital to performance on previous tasks *and* produce highly generalizable representations. Research into properties of learned representations and projection head of networks trained by contrastive loss has shown that over-parameterized (and sufficiently wide) neural networks learn embeddings with redundancy (Doimo et al., 2022; Gupta et al., 2022; Jing et al., 2021). Specifically, the vector space where the contrastive loss is defined is believed to suffer from a dimensional collapse problem (Gupta et al., 2022; Jing et al., 2021), i.e., the produced embeddings are in a lower-dimensional subspace of their nominal dimensionality. While this has been identified as an inefficiency in the normal supervised learning setting (Gupta et al., 2022; Jing et al., 2021), it poses an opportunity for continual learning: *regularization of DNN outputs can be defined only on parts of the embeddings instead of their entirety*. Similar to (Doimo et al., 2022), we observe that a small subset of contrastively learned embeddings (i.e., a subset of output neurons put together) is able to replicate the performance of the entirety of embeddings on previous tasks. Moreover, we observe that different subsets of a DNN's embeddings perform differently. By sampling random subsets of a DNN's produced embeddings and evaluating it on previous and future tasks, we see that variation of performance among subsets is higher on future tasks. These observations motivated us to define loss/regularization only on a small part of the network's outputs, one chosen such that it's likely to transfer to future tasks.

To choose a highly generalizable subset, we propose to evaluate the network on the first batch of new task data (as a surrogate of unseen future data) during task boundaries. We introduce a novel process to identify the parts of the embeddings that perform best (a subset), and a novel loss to regularize this high-performing subset. We then introduce a novel extension of the Excitation Backprop (Zhang et al., 2018) to measure the contribution of each network parameter in producing the identified subset and a novel method to modulate the gradients based on this contribution. We will describe the details of our methods in the methods section, followed by the experimental setup and results. In ablations studies, we will justify our design choices and we conclude with a discussion of methods used and how they can be improved in the future.

## 2 METHODS

We will use Co$^2$L (Cha et al., 2021) as our baseline and briefly overview its methods. We will then build our proposed methods around it.

**Continual Learning Settings:** Continual learning involves training a model on a sequence of tasks $\mathcal{T}_1, \mathcal{T}_2, ..., \mathcal{T}_n$. Each task is defined by its corresponding input and target datasets $(X_t, Y_t)$ which are drawn from a task-specific distribution $D_t$. Continual learning is mainly studied in three settings: task-incremental, class-incremental, and domain-incremental. In the task-incremental setting, the samples in each task are accompanied by a task identifier. As a result, during inference, a model can use the task identifier to drastically limit target predictions. In the class and domain incremental setting, there is no knowledge of the task identifier at inference time and the targets to predict can be among all classes seen so far by the model. While the set of target classes in each task is disjoint in the task and class incremental settings, the set of classes remains the same in the domain-incremental setting ($Y_t$ distribution stays the same while $X_t$ distribution varies).

**Contrastive Learning and Co$^2$L Overview:** Supervised contrastive learning (Khosla et al., 2020) generally involves a feature extractor mapping input samples to representations and a projection head mapping representations to embeddings. Formally, denoting a feature extractor parameterized by $\theta$ as $f_\theta$, representations by $\mathbf{r}$, projection head parameterized by $\psi$ as $g_\psi$, and embeddings by $\mathbf{e}$, supervised contrastive learning (Khosla et al., 2020) and Co$^2$L (Cha et al., 2021) augment each input sample $x$ in the minibatch twice to get $\hat{x}_1$ and $\hat{x}_2$, known as views. Representations are generated by passing the views $\hat{x}_1$ and $\hat{x}_2$ to the feature extractor ($\mathbf{r}_1 = f_\theta(\hat{x}_1), \mathbf{r}_2 = f_\theta(\hat{x}_2)$). Embeddings are then produced by passing the extracted representations to the projection head ($\mathbf{e}_1 = g_\psi(\mathbf{r}_1), \mathbf{e}_2 = g_\psi(\mathbf{r}_2)$). Both embeddings and representations are normalized to be of unit length ($|\mathbf{r}| = 1, |\mathbf{e}| = 1$). A contrastive loss is then defined on these embeddings and used to train the network. In the specific case of Co$^2$L (Cha et al., 2021), this loss is called the Asynchronous Supervised Contrastive Loss (Async SupCon) and defined as:

$$\mathcal{L}_{\text{Async}}^{\text{SupCon}} = \sum_{i \in S} \frac{-1}{|\mathcal{P}_i|} \sum_{j \in \mathcal{P}_i} \log \left( \frac{\exp(\mathbf{e}_i \cdot \mathbf{e}_j / \tau)}{\sum_{k \neq i} \exp(\mathbf{e}_i \cdot \mathbf{e}_k / \tau)} \right)$$

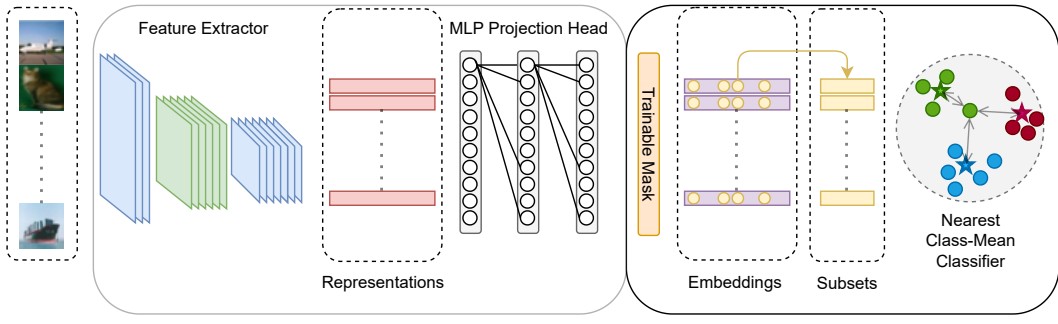

Figure 1: During task boundaries, the feature importance module is added on top of the embeddings to identify the salient subset. The mask marking the salient subset is trained based on a nearest class-mean classifier and regularized to be minimal (criterion 3).

where $S$ includes the index of views from the current task, $\mathcal{P}_i$ holds the index of views in the minibatch belonging to the same class as the $i$th view $\hat{x}_i$ except for $\hat{x}_i$ itself, $\tau$ is a temperature hyperparameter, and $\mathbf{e}_i$ is the embedding of the $i$th view. To facilitate comparison with previous work, we also employ the Async SupCon loss to train the network.

To adapt supervised contrastive learning to solve a continual learning problem and mitigate forgetting, Co$^2$L (Cha et al., 2021) uses an Instance-wise relation distillation loss (IRD). IRD computes a similarity matrix by measuring the similarity of each view with other views in the minibatch (one row) for both the old model (snapshot of the current model taken when training on the current task started and parameterized by $\omega$) and the model currently being trained (parameterized by $\theta$). The resulting two similarity matrices are then regularized to be similar to each other. Formally, the similarity of views $\hat{x}_i$ and $\hat{x}_j$ is computed as follows:

$$R_{\theta,\eta_1}[i,j] = \mathrm{Sim}(\hat{x}_i, \hat{x}_j, \eta_1, \theta) = \frac{\exp(\mathbf{e}_i \cdot \mathbf{e}_j / \eta_1)}{\sum_{k \neq i}^{2N} \exp(\mathbf{e}_i \cdot \mathbf{e}_k / \eta_1)} \tag{1}$$

where $[i,j]$ is used to denote the element in the $i$th row and $j$th column of the pairwise similarity matrix, $\mathrm{Sim}$ is the similarity function, $\eta_1$ is a temperature hyperparameter, and $N$ is the number of samples in the minibatch. The IRD loss is then defined as:

$$\mathcal{L}_{\mathrm{IRD}} = \sum_{i=1}^{2N} \sum_{j=1}^{2N} -R_{\omega,\eta_2}[i,j] \cdot \log(R_{\theta,\eta_1}[i,j]) \tag{2}$$

We believe that this distillation loss is too limiting and diminishes the model's ability to learn new generalizable representations since redundant parts of the embeddings are also regularized. We will modify this distillation loss to be applied only to a subset of embeddings. This subset will be identified by our novel *feature importance module* and regularized using our novel *selective distillation loss*. Similar to rehearsal-based continual learning approaches, we will employ a small memory to store samples. The memory size will be similar to previous work for comparison and each class will be assigned an equal portion of memory. Additionally, we extend the Excitation Backprop (Zhang et al., 2018) framework to measure the contribution of individual network parameters in producing the identified salient subset as their salience. Our novel *gradient modulation* method will then use these salience values to decrease the gradients for salient network parameters. In the following sections, we will introduce the building blocks for these methods in more detail.

**Proposed Salient Subset Selection:** To improve the IRD loss (Cha et al., 2021) we try to identify a subset of the embeddings that satisfies the following criteria and refer to it as *salient*:

1. Transfers better to unseen data compared to other subsets (based on performance on unseen tasks),
2. Contains more information about past tasks compared to other subsets (based on performance on previous tasks),

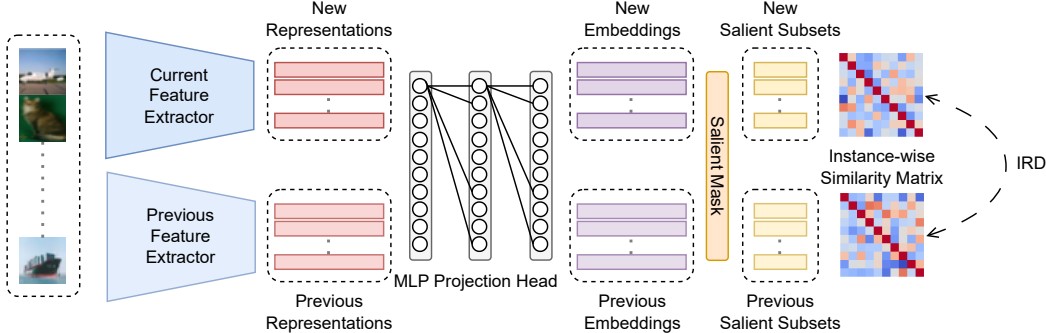

Figure 2: The Instance-wise Relation Distillation Loss is applied only to a subset of embeddings deemed to be salient by the feature importance module.

3.  Is minimal, i.e., does not have a subset that performs as well on previous and future tasks.

In the general continual learning formulation, criterion 1 and 2 can not be evaluated for a subset since we can not store all of the samples seen so far and future samples are yet to be seen. In a task boundary, however, we can use the samples stored in the memory $\mathcal{M}$ as a surrogate for previous tasks and the first batch of new task data $\mathcal{B}_t$ (before the model is trained on it) as a surrogate for future tasks. We can create a dataset $\mathcal{D}_{\text{SRS}}$ to use for finding the salient subset. This dataset can be formed by combining $\mathcal{M}$ and $\mathcal{B}_t$ (the *combined* setting, $\mathcal{D}_{\text{SRS}} = \mathcal{M} \cup \mathcal{B}_t$), or using memory samples only (the *onlypast* setting, $\mathcal{D}_{\text{SRS}} = \mathcal{M}$), or using the first batch only (the *onlycurrent* setting, $\mathcal{D}_{\text{SRS}} = \mathcal{B}_t$). These options will be assessed in the ablations studies.

Identifying a salient subset of the embeddings is essentially a search problem. Here, for simplicity and speed, we adopt an approach similar to a previous work called Neural Similarity Learning (Liu et al., 2019). Using the same notation as before, let $\mathbf{s}$ be a vector the same size as $\mathbf{e}$, $\sigma$ denote the sigmoid function, and $h_{\mathbf{s}}()$ a Nearest Class Mean Classifier (NCMC) parameterized by $\mathbf{s}$, taking in embeddings and assigning them to the class with the nearest mean embedding. Before training $h_{\mathbf{s}}()$ we need to compute class means. Let $\mathcal{D}_c$ denote samples in dataset $\mathcal{D}_{\text{SRS}}$ belonging to class $c$, then the mean of class $c$ (denoted by $\mathbf{m}_c$) can be computed as:

$$\mathbf{m}_c = \frac{\sum_{(x,t)\in\mathcal{D}_c} g_\psi(f_\theta(x))}{|\mathcal{D}_c|}$$

where $x$ is a view of an input sample and $t$ is the class it belongs to.

To train $h_{\mathbf{s}}$ we will minimize the following loss function:

$$\ell_{\mathbf{s}}(\mathcal{D}_{\text{SRS}}) = \frac{\sum_{(x,t)\in\mathcal{D}_{\text{SRS}}} \frac{\mathbf{e}\odot\sigma(\mathbf{s})}{|\mathbf{e}\odot\sigma(\mathbf{s})|} \cdot \frac{\mathbf{m}_t\odot\sigma(\mathbf{s})}{|\mathbf{m}_t\odot\sigma(\mathbf{s})|}}{|\mathcal{D}_{\text{SRS}}|} + \lambda|\mathbf{s}|_1$$

where $\odot$ denotes the element-wise product. An $\ell_1$ norm loss (with $\lambda$ as a hyperparameter controlling strength) is added on $\mathbf{s}$ to ensure it marks a minimal subset (criterion 3). The size of this subset can vary based on the redundancy of embeddings. After training the NCMC for a number of randomly initialized mask vectors and selecting the best-performing mask, $\hat{\mathbf{s}} = \sigma(\mathbf{s})$ can be used to identify which parts of the embedding should be regularized. Compared to Neural Similarity Learning (Liu et al., 2019), multiplying $\hat{\mathbf{s}}$ to the output of the encoder and class means is similar to implementing a weighted dot product, weights that are used to mask parts of the encoder's output in our case.

**Proposed Selective Distillation:** *Selective Distillation* modifies the IRD loss to be applied only on the salient subset of the produced embeddings. By applying this distillation loss selectively, we try to keep only parts of the embeddings that are salient, offer the model more flexibility in learning the new task, and encourage transfer and generalizability. Our proposed variant of IRD forms new embeddings $\hat{\mathbf{e}}$ by taking parts of the embeddings where the mask vector $\hat{\mathbf{s}}$ is above a threshold (here 0.5 is simply chosen):

$$\hat{\mathbf{e}} = \frac{\mathbf{e}[\hat{\mathbf{s}} \leq 0.5]}{|\mathbf{e}[\hat{\mathbf{s}} \leq 0.5]|}$$

The instance-wise similarity matrix (equation 1) is then computed using the new embeddings:

$$R_{\theta,\eta_1}[i,j] = \text{Sim}(\hat{x}_i, \hat{x}_j, \eta_1, \theta) = \frac{\exp(\hat{\mathbf{e}}_i \cdot \hat{\mathbf{e}}_j / \eta_1)}{\sum_{k \neq i}^{2N} \exp(\hat{\mathbf{e}}_i \cdot \hat{\mathbf{e}}_k / \eta_1)} \tag{3}$$

IRD loss (equation 2) is then calculated using the new instance-wise similarity matrices.

**Salient Parameter Selection:** After identifying the salient subset, the computed salience can be passed down using a novel extension of excitation backprop (EB) (Zhang et al., 2018). Normally, EB is a method to attribute the *activation* of a model's output neurons to its *input*. Our goal, however, is different: we want to attribute the *performance* of the salient neurons in the output layer to individual network *parameters* given a batch of data samples.

Assuming a simple neuron computes $a_i^{l+1} = \phi(\sum_j w_{j,i}^l a_j^l)$ where $a_i^{l+1}$ is the activation value of the $i$th neuron in the $(l+1)$th layer, $w_{j,i}^l$ the weight connecting the $j$th neuron in the $l$th layer to the $i$th neuron in the $(l+1)$th layer, and $\phi$ is a non-linear activation function, EB defines salience of the activation of a neuron as its *winning probability* $P(a)$. To compute the salience, it uses Marginal Winning Probability (MWP) of a neuron given neurons in the upper layer:

$$P(a_j) = \sum_{a_i \in \mathcal{P}_j} P(a_j | a_i) P(a_i) \tag{4}$$

$\mathcal{P}_j$ denotes the neurons in the layer above (closer to output) of $a_j$. Given certain assumptions (see (Zhang et al., 2018), holds when ReLU activation function is being used), the MWP for a neuron $a_j^l$ can be computed based on the salience of neurons $a_i^{l+1}$ in the upper layer:

$$P(a_j^l | a_i^{l+1}) = \begin{cases} Z_i a_j^l w_{j,i}^l & \text{if } w_{j,i}^l \geq 0, \\ 0 & \text{otherwise.} \end{cases} \tag{5}$$

Where $Z_i$ is a normalization factor and is equal to $\frac{1}{\sum_{j, w_{j,i}^l \geq 0} a_j^l w_{j,i}^l}$. Using MWPs computed from (5), the salience of each neuron can be computed in the top-down order based on (4).

To attribute the salience of output neurons to the model's *parameters*, similar to (Kolouri et al., 2019) we first employ EB to compute salience for activation maps in each layer. Next, similar to Oja's rule (Oja, 1982), the salience of each network weight can be computed using the salience of its two ends:

$$\gamma(w_{i,j}^l) = \sqrt{P(a_i^l) P(a_j^{l+1})}$$

where $\gamma$ represents salience. The output of this *salient parameter selection* process is essentially a salience value for each network parameter. These salience values will be used in the next step to modulate gradients.

**Gradient Modulation:** Inspired by neuromodulation processes in the brain where the plasticity of neurons can change based on the task at hand (Mei et al., 2022), we attempt to limit change in network weights that are deemed to be salient. In the domain of neural networks, that translates to modifying the gradients so that the more salient a network weight is, the smaller the gradient is modified to be. To achieve this, we modify the gradients as follows:

$$d_w = d_w \times (1 - \min(1, \gamma(w))) \tag{6}$$

where $d_w$ denotes the gradient with respect to parameter $w$. This process aims to guide the network during training by shifting its focus on learning the task at hand using parameters that did not contribute to the performance of the salient subset.

## 3 RESULTS

**Evaluating Random Subsets:** Motivating our work, we test the hypothesis of whether using the first batch of new task data during task boundaries is beneficial. Specifically, we want to see whether subsets of network-generated embeddings have the same discrimination power regarding past versus future tasks. At each task boundary, we extract 10 random neurons of the network-generated embedding to form a subset. Using only the selected subset, we first train a linear classifier to discriminate

Table 1: Comparing our proposed methods with published methods. All of our proposed methods were run using the onlycurrent setting of salient subset selection. Accuracy of the proposed methods was obtained by averaging across 5 independent trials. The highest accuracy marked in bold. '-' denotes settings where evaluation was impossible due to incompatibility or intractable training processes. Previous results listed are based on (Cha et al., 2021). Data are presented as mean (SD).

| Memory Size | Dataset | SplitCIFAR10 | | SplitTinyImageNet | | R-MNIST |
|---|---|---|---|---|---|---|
| | Scenario | Class-IL | Task-IL | Class-IL | Task-IL | Domain-IL |
| 200 | ER | 44.79 (1.86) | 91.19 (0.94) | 8.49 (0.16) | 38.17 (2.00) | 93.53 (1.15) |
| | GEM | 25.54 (0.76) | 90.44 (0.94) | - | - | 89.86 (1.23) |
| | A-GEM | 20.04 (0.34) | 83.88 (1.49) | 8.07 (0.08) | 22.77 (0.03) | 89.03 (2.76) |
| | iCaRL | 49.02 (3.20) | 88.99 (2.13) | 7.53 (0.79) | 28.19 (1.47) | - |
| | FDR | 30.91 (2.74) | 91.01 (0.68) | 8.70 (0.19) | 40.36 (0.68) | 93.71 (1.51) |
| | GSS | 39.07 (5.59) | 88.80 (2.89) | - | - | 87.10 (7.23) |
| | HAL | 32.36 (2.70) | 82.51 (3.20) | - | - | 89.40 (2.50) |
| | DER | 61.93 (1.79) | 91.40 (0.92) | 11.87 (0.78) | 40.22 (0.67) | 96.43 (0.59) |
| | DER++ | 64.88 (1.17) | 91.92 (0.60) | 10.96 (1.17) | 40.87 (1.16) | 95.98 (1.06) |
| | Co$^2$L | 65.57 (1.37) | 93.43 (0.78) | 13.88 (0.40) | 42.37 (0.74) | 97.90 (1.92) |
| | **SD (ours)** | **73.72 (0.52)** | **96.10 (0.09)** | **16.02 (0.39)** | **44.07 (0.66)** | **98.80 (0.26)** |
| | **GM (ours)** | 71.30 (1.15) | 95.84 (0.25) | 12.46 (0.43) | 38.33 (0.90) | 97.29 (0.59) |
| | **SD + GM (ours)** | 70.64 (0.98) | 95.28 (0.46) | 12.93 (0.55) | 38.47 (0.68) | 96.68 (0.55) |
| 500 | ER | 57.75 (0.27) | 93.61 (0.27) | 9.99 (0.29) | 48.64 (0.46) | 94.89 (0.95) |
| | GEM | 26.20 (1.26) | 92.16 (0.64) | - | - | 92.55 (0.85) |
| | A-GEM | 22.67 (0.57) | 89.48 (1.45) | 8.06 (0.04) | 25.33 (0.49) | 89.04 (7.01) |
| | iCaRL | 47.55 (3.95) | 88.22 (2.62) | 9.38 (1.53) | 31.55 (3.27) | - |
| | FDR | 28.71 (3.23) | 93.29 (0.59) | 10.54 (0.21) | 49.88 (0.71) | 95.48 (0.68) |
| | GSS | 49.73 (4.78) | 91.02 (1.57) | - | - | 89.38 (3.12) |
| | HAL | 41.79 (4.46) | 84.54 (2.36) | - | - | 92.35 (0.81) |
| | DER | 70.51 (1.67) | 93.40 (0.39) | 17.75 (1.14) | 51.78 (0.88) | 97.57 (1.47) |
| | DER++ | 72.70 (1.36) | 93.88 (0.50) | 19.38 (1.41) | 51.91 (0.68) | 97.54 (0.43) |
| | Co$^2$L | 74.26 (0.77) | 95.90 (0.26) | 20.12 (0.42) | **53.04 (0.69)** | **98.65 (0.31)** |
| | **SD (ours)** | **76.49 (0.63)** | **96.39 (0.20)** | **21.49 (0.50)** | 52.69 (0.45) | 98.43 (0.38) |
| | **GM (ours)** | 74.63 (0.95) | 96.15 (0.14) | 17.54 (0.44) | 48.21 (0.54) | 97.17 (0.50) |
| | **SD + GM (ours)** | 73.82 (0.42) | 95.67 (0.14) | 19.01 (0.31) | 48.06 (0.71) | 96.49 (1.15) |

between classes in the entirety of past tasks' data and then train another linear classifier using the same subset of neurons to discriminate between classes in the entirety of unseen tasks' data. We repeat this process 100 times and record the accuracy of the selected subset on past and unseen task data. We then compute the mean and variance of subset accuracy among these 100 subsets. We observed a higher variance when evaluating on unseen tasks (see appendix A.1 for details) suggesting that generalizability of subsets varies more than their captured knowledge of past tasks.

**Proposed Method Results:** To allow comparison with previous results (Cha et al., 2021), we conduct experiments in the task-incremental, class-incremental, and domain-incremental settings on CIFAR-10 (Krizhevsky et al., 2009), TinyImageNet (Le & Yang, 2015), and R-MNIST datasets (Lopez-Paz & Ranzato, 2017) (for experimental setup details see A.2). We compare our results with rehearsal-based continual learning methods including Co$^2$L (Cha et al., 2021), ER (Riemer et al., 2018), iCaRL (Rebuffi et al., 2017), GEM (Lopez-Paz & Ranzato, 2017), A-GEM (Chaudhry et al., 2018), FDR (Benjamin et al., 2018), GSS (Aljundi et al., 2019), HAL (Chaudhry et al., 2021), DER (Buzzega et al., 2020), and DER++ (Buzzega et al., 2020). A low (200 samples) and a high memory setting (500 samples) were considered. Results are average test-set classification accuracy on all seen classes at the end of training.

We compare the results of our proposed methods to previous work in table 1. We will use SD to refer to our selective distillation method, GM to refer to our gradient modulation method when the IRD loss is applied the same as (Cha et al., 2021) and not selectively as in SD, and SD + GM to refer to the use of both gradient modulation and selective distillation at the same time. SD improves upon baselines and state-of-the-art for both task and class-incremental settings on the SplitCIFAR10 and SplitTinyImageNet datasets. It is also superior to previous work in the domain-incremental setting on the R-MNIST dataset when a small memory is being employed. GM and SD+GM also improved

Table 2: Comparison of SD performance for different settings of the salient subset selection process. The onlycurrent setting uses the first batch of the new task, onlypast uses samples in the memory, and combined uses both for identification of the salient subset in embeddings. Adding the first batch of new task data improves SD performance in virtually all scenarios and datasets. Five independent experiments were conducted for each case to report the mean and variance. A memory buffer of 500 samples was used in all experiments.

| Dataset | SplitCIFAR10 | | SplitTinyImageNet | |
|---|---|---|---|---|
| Setting | Class-IL | Task-IL | Class-IL | Task-IL |
| onlypast | 75.33 (0.53) | 96.28 (0.15) | 21.42 (0.25) | 52.64 (0.55) |
| combined | 75.20 (0.88) | 96.29 (0.17) | 22.07 (0.37) | 52.78 (0.35) |
| onlycurrent | 76.49 (0.63) | 96.39 (0.20) | 21.49 (0.50) | 52.69 (0.45) |

state-of-the-art on the SplitCIFAR10 dataset but did not surpass SD. A discussion of GM is provided in appendix A.3. These results show that SD can successfully mitigate forgetting while freeing up the remaining parts of the model to learn new tasks. In the next section, we will analyze the choice of selecting the salient subset only based on the new batch of data rather than memory samples or combined. We will also go over the effect of embedding size for our method (SD) as it depends on the redundant units in the output of the projection head (embeddings).

## 4 ABLATION STUDIES

**Identifying the Salient Subset of Embeddings, onlycurrent, onlypast, or combined:** Although the proposed methods outperformed published methods, it was unclear which parts of our approach contributed to the performance gain. In salient subset selection, three settings were used to generate $\mathcal{D}_{\text{SRS}}$. The salient subset was then chosen based on the classification performance of subsets on $\mathcal{D}_{\text{SRS}}$. Initially, we hypothesized that including the first batch of new task data would help the salient subset selection identify parts of the embeddings that not only perform well on previous tasks but also generalize well to unseen tasks. To examine this hypothesis, we conducted experiments (Results in Table 2) on the SplitTinyImageNet and SplitCIFAR10 datasets in these three settings using the selective distillation method.

For the SplitCIFAR10 dataset, the onlycurrent setting where $\mathcal{D}_{\text{SRS}} = \mathcal{B}_t$ outperformed the onlypast and combined settings. The lower performance of the combined setting compared to the onlypast setting can be explained by the low number of classes in the CIFAR10 dataset. Added memory samples in the $\mathcal{D}_{\text{SRS}}$ dataset may be misleading as a significant portion of memory samples will belong to the task the model was just trained on. The performance of various parts of embeddings on the previous task may be less informative as it measures neither resilience to forgetting nor generalizability. Experimenting on the SplitTinyImageNet dataset, we observed that both the onlycurrent and combined settings outperformed the onlypast setting. It is worth emphasizing that the onlycurrent setting outperformed the onlypast setting on both datasets and continual learning scenarios, suggesting that using a batch of new task data may be useful for identifying the salient subset. Also note that all accuracies listed in Table 2 were higher than previous state-of-the-art results (Cha et al., 2021), demonstrating that while changing the default continual learning protocol to use the first batch of new task data may improve model performance to some extent, the main performance gains were results of the selective distillation (SD) method itself.

**The Effect of the Embedding Size:** In our first experiments, we noticed that SD outperformed $\text{Co}^2\text{L}$ (Cha et al., 2021) on all datasets except for SplitTinyImageNet. Our hypothesis was that SD relied on redundancy in the embeddings and when the generated embeddings were dense, it was reasonable to apply the IRD loss on entire embeddings rather than a subset. Moreover, since in contrastive learning the projection head is discarded after training and is generally small (MLP, 512 hidden units, 128 output units in $\text{Co}^2\text{L}$), increasing embedding size to induce redundancy comes with virtually no computational cost, especially at inference time. To test our hypothesis, we compared SD to $\text{Co}^2\text{L}$ (Cha et al., 2021) with different embedding sizes on the SplitCIFAR10 and SplitTinyImageNet datasets. For SplitCIFAR10, the embeddings seemed to be dense when the embedding size was around 16 and started to involve some redundancy starting from 32 units in the output (figure

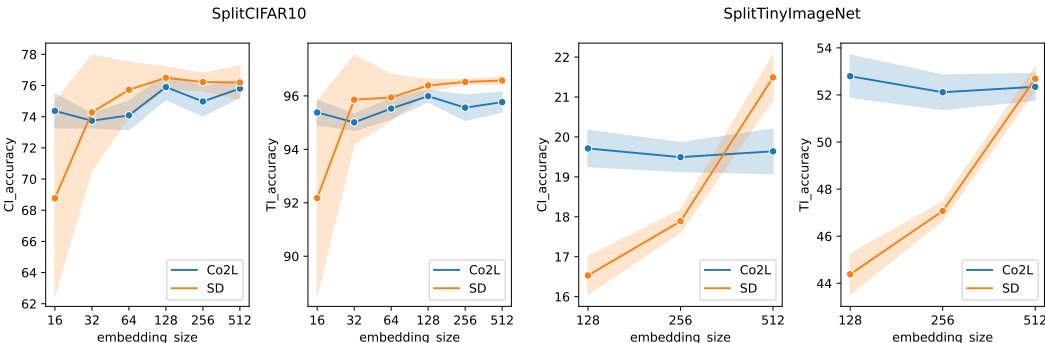

Figure 3: Comparing SD (ours) and Co$^2$L (Cha et al., 2021) using different embeddings sizes on the SplitCIFAR10 and SplitImagenNet datasets. The memory size is the same (500) for both methods. Shading depicts standard deviation. Increasing embedding size and redundancy benefits SD on both datasets.

3 left). As we increased the embedding size starting with 32 units, we noticed that SD consistently outperforms Co$^2$L (Cha et al., 2021).

When testing our hypothesis on the SplitTinyImageNet dataset (which is generally more difficult to solve with 200 classes), we noticed that embeddings appeared to be dense until an embedding size of 256 and SD was unable to outperform Co$^2$L (Cha et al., 2021). However, with an embedding size of 512, redundancy began to materialize in embeddings and SD achieved higher task- and class-incremental accuracy (figure 3 right). We did not increase the embedding size further as it would have gotten larger than the hidden layer's size and could have caused complications unrelated to this ablation study. Overall, these results showed that as the embedding size grows larger, SD can leverage the increased redundancy and improve continual learning performance in both task and class-incremental settings.

## 5    CONCLUSION

Inspired by Event Models, we proposed to look differently at the continual learning setting and focus on task boundaries. We hypothesized that the first batch of new task data could be used to identify parts of the neural network that enable generalization to unseen tasks. Observing the redundancy-inducing effects of the contrastive loss on embeddings, we first introduced a salient subset selection process where a subset performing similarly to the entirety of embeddings was identified. Secondly, we proposed a selective distillation method that regularized only the salient parts of the embeddings. Thirdly, we introduced an attribution method that assigned salience to network parameters based on their contribution to the computation of the salient subset. Fourthly, we proposed a gradient modulation method that modified gradients according to the salience of parameters. Our methods did not increase parameters linearly with the number of tasks or assume that additional memory was available in the form of a second snapshot of the model or more samples in the memory. Moreover, in alignment with our hypothesis, the selective distillation method was able to leverage redundancy in embeddings and demonstrated superior performance when compared to previous work. Our analysis suggested that research into the properties of projection heads in representation learning can induce more redundancy within the network and open new doors to challenges in continual learning.

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

# A  APPENDIX

## A.1  VARIANCE IN SUBSET ACCURACY

For the purposes of this analysis only, we will deviate from the standard continual learning protocol. Note that our main proposed methods follow the standard continual learning protocol. Training a linear classifier on previous and upcoming task data using 100 random embedding subsets of size 10 we observed that the performance of different random subsets has a noticeable variation on both past and future tasks, but the variance is generally higher on future tasks (table 3). Taking accuracy on future tasks as an indicator for generalizability of a subset, this also shows that not all subsets are equal in terms of how generalizable they are, and thus regularizing parts of the embeddings that do not transfer well is limiting the network's ability to learn new tasks. These results support our decision to include the first batch of new task data to identify the salient subset.

Table 3: Mean (std) of subset accuracy on previous and upcoming tasks are evaluated at each task boundary. The results are computed based on three independent trials on the SplitCIFAR10 dataset. The standard deviation for these results is calculated based on trials.

|  | Task 1 | Task 2 | Task 3 | Task 4 |
|---|---|---|---|---|
| Mean subset accuracy on past tasks | 99.18 (0.06) | 74.19 (0.34) | 63.09 (4.35) | 54.08 (0.97) |
| Mean subset accuracy on future tasks | 30.50 (0.53) | 38.65 (0.62) | 60.56 (0.40) | 76.81 (0.69) |
| Std of subset accuracy on past tasks | 0.23 (0.01) | **3.75 (0.10)** | 2.70 (0.40) | 2.92 (0.09) |
| Std of subset accuracy on future tasks | **2.10 (0.03)** | 2.54 (0.08) | **3.82 (0.42)** | **4.99 (0.45)** |

## A.2  EXPERIMENTAL DETAILS

We conduct experiments in three common continual learning scenarios: Task-Incremental (Task-IL), Class-Incremental (Class-IL), and Domain-Incremental (Domain-IL). For class and task-incremental settings, CIFAR-10 (Krizhevsky et al., 2009) and TinyImageNet (Le & Yang, 2015) datasets were used, while for the domain-incremental setting, we used Rotational MNIST (R-MNIST) (Lopez-Paz & Ranzato, 2017). CIFAR-10 and TinyImageNet will be divided across classes into 5 and 10 sub-datasets to create SplitCIFAR10 and SplitTinyImageNet respectively. Each task will then be to solve an image classification task on 2 classes for SplitCIFAR10 and 20 classes for SplitTinyImageNet. The order of classes is the same across experiments. The R-MNIST dataset will consist of 20 tasks, where for each task the MNIST (LeCun et al., 1998) dataset is rotated using a random degree in the range of $[0, \pi)$ (uniformly sampled). Similar to (Cha et al., 2021), when training on R-MNIST, the same digits rotated by a random degree will be treated as different classes in the Async SupCon loss. The implementation for this work is based on the implementation of (Cha et al., 2021). Unless otherwise stated, all choices of optimizer, architecture, and hyperparameters were kept the same.

For training on SplitCIFAR10 and SplitTinyImageNet, we use the ResNet-18 (He et al., 2016) architecture while for R-MNIST, the same smaller architecture as in (Cha et al., 2021) is employed for comparison. A two-layer linear network is used for the projection head. Importantly, we increase the embedding size (output of projection head) for the SplitTinyImageNet dataset. We have explained this design choice in the ablation study 4. Evaluation is according to contrastive learning framework (Cha et al., 2021; Khosla et al., 2020) which trains a classifier on top of the encoder using last task samples and samples in the memory (as if the classifier was trained immediately after learning a task, according to samples available at the time).

## A.3  DISCUSSION ON GRADIENT MODULATION

Neuromodulation-inspired mechanisms have enabled continual adaptation in a wide range of tasks, including navigation (Vecoven et al., 2020; Mei et al., 2023), language modeling Miconi et al. (2020), and image classification (Daram et al., 2020). Similarly, our approach used saliency information to modify the gradients for parameters that are identified as salient. While our Gradient modulation surpassed state-of-the-art performance on the SplitCIFAR10 dataset with or without selective distillation, it did not perform better than SD. Although it may seem that GM is not a promising technique for continual learning, it is worth noting that our extension of the Excitation

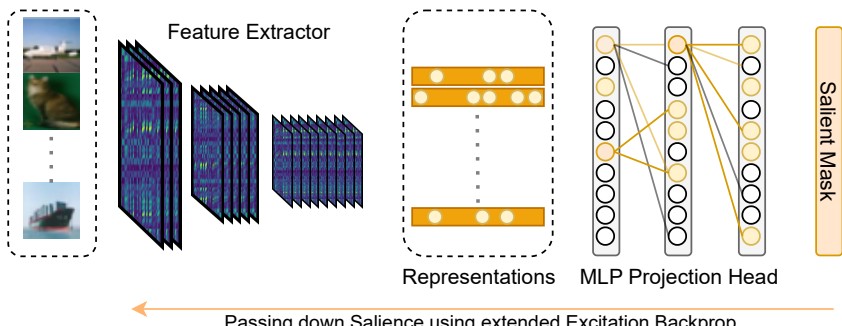

Figure 4: Using the mask vector $\hat{\mathbf{s}}$ to pass down salience to each network weight using our extended version of Excitation Backprop (Zhang et al., 2018). The computed parameter importance is then used to modulate gradients.

Backprop (Zhang et al., 2018) provides useful saliency and attribution information. It can compute for each network parameter a salience value describing its contribution in forming a specific subset in the embeddings. We used the parameter salience information produced by this framework to modulate gradients to encourage learning new tasks using parts of the network that did not seem to contribute to salient features in the embeddings. However, the parameter salience information can be used in many different ways, e.g., identifying parameters to regularize and preserve, finding sub-networks that are capable of performing similarly to the whole network, or finding the least salient parameters. To our knowledge, this method is the only variant of Excitation Backprop Zhang et al., 2018 that can be used in networks where the loss function is defined on representations or embeddings (representation learning). It is also worth noting that this method can assign salience based on the performance of the generated embeddings and representations, not just the activations of certain neurons. Overall, we believe GM is a multi-purpose tool with use cases that go beyond continual learning.

## A.4 DISCUSSION ON MEMORY AND COMPUTE USAGE

Following the general continual learning desiderata (Hadsell et al., 2020) we focused on using a fixed-capacity model. As a result, we did not include model-growing approaches such as Progressive Neural Networks (Rusu et al., 2016) and TAMiL (Bhat et al., 2023) in our reported results. We also did not consider multiple memory approaches where the memory usage goes further than a copy of the main model and a memory of samples. These approaches include a recent promising work called CLS-ER (Arani et al., 2022) where two exponentially averaged copies of the model are maintained. Although our approach outperforms CLS-ER on the SplitCIFAR10 dataset, we believe methods with two memory systems should be compared with each other, and single memory systems should be compared with one another for a fair comparison. A copy of the ResNet-18 architecture is typically 40 MB in size, while each image in the TinyImageDataset is about 3 KB. The low memory setting assumes access to memory is so limited that only 200 samples (one per class) can be stored. The addition of a copy of a ResNet-18 model is similar to adding more than 10,000 samples to this memory and thus gives a significant advantage compared to a method that employs one copy only.

Furthermore, empirical results in recent work (Arani et al., 2022; Bhat et al., 2023; Sarfraz et al., 2023) suggest that using an exponentially averaged model over the trajectory of learning is more robust in mitigating forgetting compared to using a static snapshot of the model from a single point in time. However, to emphasize the robustness of our methods, we decided to test them in a standalone manner and did not use this technique. We leave it to future work to combine our methods with CLS-ER (Arani et al., 2022) or ESMER (Sarfraz et al., 2023) and study the effects.

## A.5 COMPUTATIONAL COMPLEXITY OF THE GRADIENT MODULATION METHOD

The proposed GM is implemented based on Excitation Backprop (Zhang et al., 2018), computing importance for weights in addition to activations. To compute the importance of activations, first, a forward pass takes the inputs to the network and computes layer activations. EB (Zhang et al.,

2018) then performs a backward pass, computing the importance of activations from top to bottom. In each layer, EB computes raw importance values for the lower layer and then performs a mini-forward pass to normalize these importance values. We perform an additional mini-forward and a mini-backward pass to attribute the importance of activations to layer weights. This is independent of the layer type and works based on Pytorch's autograd functionality. As a result, GM performs two forward and backward passes to compute the salience of the network parameters. Note that this computation is performed only once during task boundaries and does not occur during training on a task.

