# OpenReview forum: "Look-Ahead Selective Plasticity for Continual Learning of Visual Tasks"
_ICLR.cc/2024/Conference — ICLR 2024 Conference Withdrawn Submission_

### Official Review · Reviewer_a9u9 · 2023-10-29

**Soundness:** 2 fair
**Presentation:** 3 good
**Contribution:** 2 fair
**Rating:** 5
**Confidence:** 4

**Summary:**

This paper focused on contrastive learning methods in continual learning, and proposed to leverage the first few samples of the new task to identify and retain parameters contributing most to the transfer ability of the neural network, freeing up the remaining parts of the network to learn new features. The authors claimed that this idea is inspired from event models of the brain. The proposed method achieves some improvements on relatively simple dataset.

**Strengths:**

1. The paper is basically well-written and easy to follow.

2. The idea of event models is interesting. It’s good to see the connections between task boundaries and neurological mechanisms.

**Weaknesses:**

1. I agree that the current continual learning methods focus more on stability rather than plasticity/transfer. However, I think the technical contribution is incremental and not completely novel. The proposed method can be seen as an improved version of Co$^2$L. Also, the idea of “look-ahead” new tasks has been widely discussed in recent literature, such as learning and combing the new task solution [1] [2]. These related work should be discussed and compared (at least conceptually).

2. The proposed method can only achieve marginal improvements over Co$^2$L, especially for TinyImageNet in Table 1. Also, the considered benchmarks are relatively simple in continual learning.

3. The ablation study is not very clear, and the performance differences are marginal between each baseline in Table 2 (considering the error bars).

[1] Afec: Active forgetting of negative transfer in continual learning. NeurIPS 2021.

[2] Towards better plasticity-stability trade-off in incremental learning: A simple linear connector. CVPR 2022.

**Questions:**

Please refer to the weakness.

---

> ### Author Response · Authors · 2023-11-14
>
> Thank you for taking the time to review our work.\
> Addressing weaknesses:
> 1. To our knowledge, no prior work has (1) used the redundancy of embeddings resulting from contrastive loss to define the regularization of previous knowledge on a minimal subset and (2) measured the contribution of groups of neurons in the output of the network and each individual network parameter to the generalizability of the network.\
> We appreciate your mention of AFEC [1] and [2], and we apologize that we did not include the mentioned line of research. The idea of selectively configuring the plasticity of neurons in the network appears similar, but importantly, the mentioned work’s idea of plasticity is in a limited form:\
> Let $S_{1:(t-1)}$ be the set of features each neuron computes when continually trained on tasks 1 to t-1. Also, let $S_t$ be the set of features each neuron computes when trained on task t. The mentioned line of work thinks of plasticity as selecting/combining $S_{1:(t-1)}$ with $S_t$ to have the lowest loss/highest performance on all the tasks seen so far and the new task. The set of features will then be limited to $S_{1:(t-1)} \cup S_t$ and their combinations. These approaches do not encourage learning features that work on all tasks at the same time (similar to the neurons resulting from joint training of the network on all tasks). That is not our objective and definition of plasticity. When GM does not reduce the gradient of a set of neurons because they did not contribute to the transfer ability of the network, it does not mean they will be replaced by features that perform well on the new task or it will be a mix of past and new features. The new computed feature could, in theory, work well on both tasks, when trained using contrastive loss that encourages generalizability with rehearsal. We further encourage generalizability by preserving a neuron if it contributes to the network’s transfer ability. We believe this is a fundamental difference that sets our approach apart from previous work. We do agree that the objective of finding and selecting parameters to regularize or give more freedom to is similar, but our ultimate goal of having a general continual learner that leverages redundancy and modulates the plasticity of its parameters with a focus on transfer to new tasks is different.
> 2. We believe that we have shown substantial improvements in Table 1, especially in the low-memory setting. On the lower memory setting of 200 samples and the class-incremental scenario which is considered the more difficult task, the accuracy gain is 8.15% on CIFAR10, 2.14% on TinyImageNet, and 0.9% on R-MNIST in the domain-incremental scenario. We will add CIFAR100-10split in the revision. The datasets chosen are standard datasets for rehearsal-based continual learning approaches, widely used as in [3][4][5][6][7].
> 3. We apologize that the ablation studies were not explained clearly and will explain them better in the revision. In the first ablation study, we wanted to know that how selecting the subset of output neurons to regularize impacts performance. As mentioned in “Salient Subset Selection”, this subset can be found by optimizing the performance on the first batch of new task data (onlycurrent), memory samples (onlypast), or both (combined). Table 2 shows that the onlycurrent and combined settings performed better than onlypast, noting the impact of measuring the transfer of embeddings compared to finding the subset only based on its performance on previous tasks. We also noted that our performance gains were mainly due to leveraging redundancy and finding a set of output neurons that is minimal, i.e. it does not have a subset with similar performance, and not look-ahead. This is not a drawback of the work, but an attribution of the performance gains to the proposed methods. In the second ablation study, we analyzed how increasing the embedding size to increase redundancy in output neurons improves performance for the SD method.
>
> Thank you for helping us improve our work. It is platforms like ICLR that let our methods benefit researchers in continual learning and AI and we appreciate you granting us the opportunity to present them to the AI community.
>
> [1] “Afec: Active forgetting of negative transfer in continual learning”. NeurIPS 2021\
> [2] “Towards better plasticity-stability trade-off in incremental learning: A simple linear connector.” CVPR 2022\
> [3] "Class-incremental continual learning into the extended der-verse." IEEE Transactions on Pattern Analysis and Machine Intelligence\
> [4] “Learning fast, learning slow: a general continual learning method based on complementary learning system“, ICLR 2022\
> [5] “Error Sensitivity Modulation based Experience Replay: Mitigating Abrupt Representation Drift in Continual Learning”. ICLR 2023\
> [6] “Learnability and Algorithm for Continual Learning”, ICML 2023\
> [7] “Dark Experience for General Continual Learning: a Strong, Simple Baseline”, NeurIPS 2020\

---

> > ### Comment · Reviewer_a9u9 · 2023-11-22
> > **response**
> >
> > I thank the authors for addressing my concerns. After reading other reviewers' comments, I agree that there are a number of related work and relevant ideas to be compared in this work. Therefore, I cannot be more positive at the current stage.

---

### Official Review · Reviewer_ecgj · 2023-10-30

**Soundness:** 3 good
**Presentation:** 2 fair
**Contribution:** 2 fair
**Rating:** 3
**Confidence:** 5

**Summary:**

This paper proposed a new method for continual learning which is built on top of the existing work Co2L [1]. The author leverages the first batch of the new data as a surrogate to estimate the crucial parameters of the old model which are beneficial for new tasks and also important for old tasks. The estimation is done by searching for a set of embedding that can be salient for evaluating the above criteria, and the author adapts the existing Neural Similarity Learning [2] to identify these subsets. Then the author uses the Excitation Backprop (EB) to calculate the salience of each network weight and then uses the weight to mask the distillation loss for training the model to mitigate forgetting. The author also proposed a gradient modulation to modify the gradients. Extensive experiments are conducted on standard continual learning benchmarks.


Reference:
[1] Co2l: Contrastive continual learning (ICCV 2021)
[2] Neural similarity learning (NeurIPS 2019)
[3] Top-down neural attention by excitation backdrop (IJCV 2018)

**Strengths:**

1. Overall, the paper is easy to follow. Using the look-ahead idea to estimate the importance of the model weight seems to be interesting.
2. The author provides many experiments and analyses to valid and reason about the proposed method.

**Weaknesses:**

1. The look-ahead idea is not totally new in continual learning. The author did not discuss the relationship between seminal work like "La-MAML: Look-ahead Meta-Learning for Continual Learning" (NeurIPS 2020) and the present work, where the La-MAML has already considered using the initial batch of data to adapt the gradient for continual learning, which in general is related to the author's proposed masked distillation training and gradient modulation.

2. It is unclear why the paper needs to start with contrastive continual learning, i.e., Co2L, as the starting point for developing the method. First, since Co2L was published in 2021, there are so many continual learning methods that do not use contrastive learning and still achieve state-of-the-art (SOTA) performance. What is the necessity of using Co2L as the learning objective? Is it because the proposed method can not work without Co2L?

Moreover, the author stated in Page 4 that:

"We believe that this distillation loss is too limiting and diminishes the model’s ability to learn new generalizable representations since redundant parts of the embeddings are also regularized."

Could the author provide an explicit, formal, and/or empirical analysis about why the distillation loss will have such drawbacks? Such a claim is not sound, especially when we check the results in Table 1 that the proposed method does not significantly outperform the Co2L and the Co2L even outperforms the proposed method on SplitImageNet and R-MNIST. It is hard to convince the reader that the issue mentioned by the author for Co2L is grounded.

3. The author proposed to calculate the salient estimation for each parameter and use the ResNet-18 and two-layer linear network for experiments. How will the computation complexity for this salient estimation be on Page 6? Will there be a computational bottleneck when the proposed method is applied to modern neural network architecture like ViT and Transformer?

4. The CL methods compared in the present paper are up to 2021, while there are lots of new CL methods proposed after 2021 and the author did not review them in the paper and did not even mention why the author did not choose them for comparison. Moreover, the proposed method does not even significantly outperform the Co2L, i.e., the baseline they have chosen for developing their method. All of this largely hampers the significance of the current paper.

5. Although the related work section is not required, the reviewer still suggests the author to have a related work section to comprehensively review the existing CL methods especially at least discuss the recent advance of CL methods after 2021, instead of have a lengthy Introduction section which may largely distract the attention for a reader.

**Questions:**

Please refer to the Weaknesses section for more details.

---

> ### Author Response · Authors · 2023-11-14
>
> Thank you for taking the time to review our work.\
> Addressing weaknesses:
> 1. We appreciate the reviewer bringing up La-MAML. We believe the similarity of this work to our GM method is too little to question novelty. Modifying gradients and using the first batch of data are neither the main contributions of our work nor La-MAML. It is rather how these resources are used to achieve the objectives. Our goal was to encourage generalizability and use redundancy in learned embeddings to selectively increase plasticity of neurons and our results in Table 1 show we achieved this goal. La-MAML mainly aims for an efficient way to align the gradients of the current and previous tasks. While the objective of La-MAML is approximately and indirectly similar to lowering the empirical loss on all tasks, it does not measure and thus preserve features that contribute to the generalization of the network. Learning trajectories for parameters and their alignment with gradients of the new task data is essentially a heuristic and does not inform whether the neuron will ultimately learn a feature that contributes to performance on all seen tasks. In contrast, our work directly measures the contribution of parameters to the formation of a subset of output neurons that transfers well. The computed importance values could be used for gradient alignment, regularization, or any other method to modulate plasticity, and thus is more general with more applications. Note that we included ANML [1] in our review of meta-learning approaches because it was one of the few meta-learning continual learning methods that directly focused on sparsity and leveraging redundancy to learn new tasks. Moreover, gradient alignment meta-learning methods pay little attention to when a neuron merely computes the same feature as another, and train all “clone” neurons indifferently. We did not review or compare meta-learning approaches to our work extensively due to what we summarized as “Likewise, recent meta-learning approaches like La-MAML use gradient-alignment heuristics to modulate the plasticity of parameters but pay little attention to redundancy and the contribution of parameters to generalizability, while being computationally expensive compared to other continual learning methods” in the introduction, page 2. It goes against fairness to compare a rehearsal-based method with no complicated backpropagation processing to a meta-learning method.
> 2. Contrastive learning provides (1) redundancy in the embeddings resulting from dimensional collapse and (2) representations that transfer better and forget less. (1) helped our objective of increasing plasticity in the network, and (2) helped in learning features that are more generalizable. We proposed SD to identify a subset of output neurons that transfers well and freed up the rest of the network to learn new generalizable features. We have explained (2) in the introduction, page 2 “Thus, there has been a general lack of attention regarding the transfer of continually ... contrastively learned representations transfer better and forget less, compared to learning based on the cross entropy loss. ..., we were inspired to build upon the contrastive learning framework”. We also described (1) in the “Redundancy in Contrastive Learning” sub-section. To summarize, learning representations that transfer well, forget less, and include redundant parts were the main reasons to build upon the contrastive framework.
> - The claim "We believe that this distillation loss is too limiting ..." is sound and the results in Table 1 support it. We conducted experiments on task, class, and domain-incremental scenarios, on CIFAR10, TinyImageNet, and Rotational MNIST, on two low and high memory settings. Note that the performance of SD on R-MNIST for 200 memory samples outperforms Co2L with 500 memory samples. Our method (SD) outperformed Co2L on all scenarios, benchmarks, and memory settings with the exception of Task-Incremenal TinyImageNet with 500 memory samples, which given the high std is not significant. Could you explain the kind of empirical analysis that would convince the reader that the mentioned claim is true, in addition to the experiments we have conducted?
> 3. We apologize that we did not provide the computational complexity of GM. We have added this in the appendix in the revision (see A.5).
> 4. Thank you for this suggestion. We have mentioned some of the recent work in the appendix and why we did not choose to compare our work to them (see A.4). We focused on rehearsal-based continual learning methods for visual tasks and as a result did not review works related to continual learning for large language models, diffusion models, etc.
>
> Thank you for helping us improve our work. It is platforms like ICLR that let our methods benefit researchers in continual learning and AI and we appreciate you granting us the opportunity to present them to the AI community.
>
> [1] Learning to Continually Learn. ECAI 2020

---

### Official Review · Reviewer_dRbC · 2023-10-30

**Soundness:** 3 good
**Presentation:** 2 fair
**Contribution:** 2 fair
**Rating:** 3
**Confidence:** 4

**Summary:**

Paper proposes method for relevant neuron selection within a self-supervised continual learning framework; their starting point is the Co2L method. The authors propose to learn a set neurons that are relevant for the current task performance (they compare several strategies). This learned set of representation-dimensions are then used to perform a masked instance-wise relation distillation loss. Results on a few datasets shows the method improves Co2L.

**Strengths:**

- the argument that the proposed method improves the potential plasticity by reducing regularization on redundant dimensions of methods is nice. Measuring this on the current task data is new (rather on the previous)

- the proposed method obtains decent performance gain for small memory size especially on CIFAR10.

**Weaknesses:**

- the idea to focus on the importance of neurons for future (or current) tasks is new, many methods aim to measure the importance of neurons for previous tasks. However, the final difference between these strategies is very small (see table 2), and in my opinion too small.

- I do not really like CIFAR 10 for continual learning since the tasks are really small. I would like to also see results on CIFAR100 and if possible on ImageNet-subset.

 - more results on the subset size should be added.

**Questions:**

Please address the weaknesses.

For me in Table 2, the gain with respect to CO2L are ok, but not very large, and I would really like to at least also see it on CIFAR100 /10 split. Table 2 shows that selection can work; however, it also shows that any selection works and that the results among the various ways of selecting are very small (the 'look-ahead' does not seem crucial).

minor:
- I would consider removing GM from the paper since it does not improve results.
- I'm not sure if the term 'salient' is very adequate to refer to the selected neurons.
-  number of tasks used per dataset should be clearly stated in the main paper.
- add in table 2 without using the selection as well (I think it helps, even though the numbers are Table 1)

---

> ### Author Response · Authors · 2023-11-14
>
> Thank you for taking the time to review our work.\
> Addressing the weaknesses:
> - The idea of computing the importance of neurons is indeed not new. The method to find this importance varies in previous work such as [1][2][3]. Note that we propose two methods, one focused on subsets of the output neurons (Selective Distillation (SD)), and one solving a more difficult task of finding the importance of any neuron in the network (GM). Prior to this work, we tested various theoretical ways to compute the importance for each output neuron, individually. Considering a hypersphere where all classes are perfectly dispersed with maximum mutual distance, one can compute the characteristics of the projection of a cluster of samples belonging to the same class on each axis. We observed that there is not much difference between most of these neurons according to our theoretical metrics. We also observed that the features these neurons compute can combine very well and provide high accuracy in distinguishing classes.\
> In Table 2 we show the effect of the look-ahead mechanism and the improvements it brings. We also mention that “the main performance gains were results of the selective distillation (SD) method itself” in the first ablation study. SD is not about any selection, it is about finding a subset of output neurons that matches the criteria mentioned in “Proposed Salient Subset Selection”. Most notably, table 2 shows that as long as a subset does not include redundant neurons (those that compute the same features as other neurons, give or take some noise), then performance improvements can be expected. The gains from both ideas (SD and look-ahead) have been demonstrated in Table 1. On the lower memory setting of 200 and the class-incremental scenario which is considered the more difficult task, the accuracy gain is 8.15% on CIFAR10, 2.14% on TinyImageNet, and 0.9% on R-MNIST in the domain-incremental scenario. We hope with these clarifications the reviewer acknowledges the performance improvement of our proposed method compared to state-of-the-art in this field.
> - CIFAR10 is used as a benchmark in [4,5,6,7]. It has a low number of classes and tasks, but the task size is not too small. We used it for the majority of our initial testing and we assume others reporting performance on this dataset did the same. We will conduct CIFAR100 experiments in the revision for a more complete analysis of the method to address your concern.
> - We apologize if did not make it clear in the main part of the paper. The subset size is not fixed. See "Proposed Sailent Subset Selection". This mask identifying the subset is regularized to be minimal via an $\ell_1$ norm (that encourages sparsity as well). The resulting subset is usually between 40 to 90 neurons for a 128-dimensional embedding. The $\lambda$ hyperparameter controls the regularization strength.
>
> Answering the Questions:
>
> - Look-ahead is indeed not crucial, but it does give performance gains. Note that an important observation of Co2L was that contrastive loss is able to learn representations that transfer better than the cross entropy loss and forget less. The effect of contrastive loss may have diminished what look-ahead can offer. The performance gains by the look-ahead mechanism rely mainly on the variance of subset performance (see Table 3). Depending on the problem being solved and the loss function, the variance of subset accuracy may be large, and the resulting performance gain more significant.\
> It is important to note that our main method (SD) provides substantial performance gains as in Table 1, in all continual learning scenarios and datasets we tested on.
> - GM attempts to solve the more difficult problem of finding the contribution of each individual neuron to the transfer ability of the network. It is a performance attribution method with a wide range of applications that also worked in some cases for our continual learning work. We believe the readers can use, modify, and benefit from this method, and as a result, we opt for keeping it in the paper.
>
> - We will address the rest of the “minor” concerns in the revision. Thank you for helping us improve our work. It is platforms like ICLR that let our methods benefit researchers in continual learning and AI and we appreciate you granting us the opportunity to present them to the AI community.
>
> [1] "Overcoming catastrophic forgetting in neural networks." Proceedings of the national academy of sciences.\
> [2] "Continual learning through synaptic intelligence.", ICML 2017\
> [3] “Attention-based structural plasticity”.\
> [4] “Co2l: Contrastive Continual Learning”, ICCV 2021\
> [5] “Learning fast, learning slow: a general continual learning method based on complementary learning system“, ICLR 2022\
> [6] “Error Sensitivity Modulation based Experience Replay: Mitigating Abrupt Representation Drift in Continual Learning”. ICLR 2023\
> [7] “Dark Experience for General Continual Learning: a Strong, Simple Baseline”, NeurIPS 2020\

---

> > ### Comment · Reviewer_dRbC · 2023-11-21
> > **rebuttal**
> >
> > I thank the authors for their rebuttal, I would however remain with my original recommendation.